# Differential Expression of the Apolipoprotein AI Gene in Spotnape Ponyfish (*Nuchequula nuchalis*) Inhabiting Different Salinity Ranges at the Top of the Estuary and in the Deep-Bay Area of Gwangyang Bay, South Korea

**DOI:** 10.3390/ijerph182010960

**Published:** 2021-10-19

**Authors:** Kiyun Park, Won-Seok Kim, Bohyung Choi, Ihn-Sil Kwak

**Affiliations:** 1Fisheries Science Institute, Chonnam National University, Yeosu 59626, Korea; ecoblue@hotmail.com; 2Department of Ocean Integrated Science, Chonnam National University, Yeosu 59626, Korea; csktjr123@gmail.com; 3Inland Fisheries Research Institute, National Institute of Fisheries Science, Geumsan-gun 32762, Korea; chboh1982@chonnam.ac.kr

**Keywords:** *Nuchequula nuchalis*, apolipoprotein, salinity, gene expression, estuary environment

## Abstract

Spotnape ponyfish (*Nuchequula nuchalis*) is a dominant species that is broadly distributed from estuarine to deep-bay areas, reflecting a euryhaline habitat. Apolipoprotein AI (ApoAI) is a main component of plasma lipoproteins and has crucial roles in lipid metabolism and the defense immune system. In this study, we characterized the *N. nuchalis* ApoAI gene and analyzed the expression of the ApoAI transcript in *N. nuchalis* collected at various sites in the estuary and the deep-bay area which have different salinities. Owing to the fish’s mobility, we conducted stable isotope analyses to confirm the habitat characteristics of *N. nuchalis*. Carbon and nitrogen isotope ratios (δ^13^C and δ^15^N) from *N. nuchalis* indicated different feeding sources and trophic levels in the estuarine and deep-bay habitats. The characterized *N. nuchalis* ApoAI displayed residual repeats that formed a pair of alpha helices, indicating that the protein belongs to the apolipoprotein family. In the phylogenetic analysis, there was no sister group of *N. nuchalis* ApoAI among the large clades of fish species. The transcriptional expression level of ApoAI was higher in *N. nuchalis* inhabiting the deep-bay area with a high salinity (over 31 psu) than in *N. nuchalis* inhabiting the top of the estuary with a low salinity (6~15 psu). In addition, the expression patterns of *N. nuchalis* ApoAI were positively correlated with environmental factors (transparency, pH, TC, and TIC) in the high salinity area. These results suggest that ApoAI gene expression can reflect habitat characteristics of *N. nuchalis* which traverse broad salinity ranges and is associated with functional roles of osmoregulation and lipid metabolism for fish growth and development.

## 1. Introduction

Gwangyang Bay is a typical semiclosed inner bay located on the southern coast of Korea; it is influenced by the Tsushima Current, a tributary of the Kuroshio Current [1]. Freshwater from the long Seomjin River (drainage area, 4896 km^2^) flows into the northeast section of Gwangyang Bay, and the annual mean freshwater discharge into the bay from the Seomjin River is 2298 × 10^6^ m^3^ yr^−1^ [2]. The four contiguous cities of Gwangyang, Suncheon, Yeosu, and Hadong-gun are located near the bay. The bay is surrounded by paddy fields and industrial facilities, such as the Yeocheon National Industrial Complex in the south and the Gwangyang steel mill in the north. The bay extends over 450 km^2^ from the estuary to the outer bay and has a high economic productivity from fishery resources and biological diversity [3,4]. In 2018, the fish populations in Gwangyang bay were reported to include 27 fish species with high species richness and diversity (H’ = 2.35) [5]. In the outer region of the bay, the predominant fish are Spotnape ponyfish (*Nuchequula nuchalis* (Temminck and Schlegel, 1845); >40%) and silver white croaker (*Pennahia argentata* (Houttuyn, 1782); >20%) [3]. *N. nuchalis* is a small, scaleless marine fish of approximately 10–14 cm in length and belongs to the Leiognathidae family of the order Perciformes; it inhabits the shallow waters of southern Korea, Japan, the East China Sea, and the Pacific Ocean [6,7,8]. *N. nuchalis* is a carnivore that mainly consumes crabs and copepod larvae and has the plastic feeding strategy from zooplanktonic (estuarine habitat) to epibenthic (deep-bay habitat) feeder during the migration in the Seomjin estuary and Gwangyang Bay [9,10]. It is difficult to externally distinguish males from females of this fish species. *N. nuchalis* is an important environmental aquatic species owing to its high nutritional and economic values. However, no genomic information is available for this species.

Seawater salinity varies seasonally in coastal areas owing to tides, surface runoff, and rainfall [11] salinity is one of the most pivotal environmental factors affecting fish survival, as it can directly affect growth, embryo hatchability, development, feeding, and digestion [12]. In marine organisms, salinity changes also affect molecular processes including ion transport, energy metabolism, antioxidant system, and signal transduction [13,14,15,16]. Considerable metabolic energy is required to regulate osmotic pressure and ionic balance in response to salinity changes in the aquatic environment [16,17].

Apolipoprotein AI (ApoAI) is the most abundant and key component of plasma lipoproteins [18,19]. ApoAI participates in lipid transport and osmotic homeostasis of an organism, which are among the most important functions of plasma proteins [20,21]. Plasma proteins, such as ApoAI, play a crucial role in maintaining tissue fluid balance because they are preserved within capillaries and create colloid osmotic pressure [22]. ApoAI is involved in antimicrobial activity in response to bacterial and viral challenges and play important roles in innate immunity [21,23,24]. In addition, anti-obesity and anti-atherogenic functions of apolipoprotein were reported in anchovy (*Engraulis encrasicolus* (Linnaeus, 1758)) [25,26]. Despite their importance, the ApoAI gene has not been systematically and widely characterized in many fish species. The goal of this study was to identify the changes in ApoAI gene expression in response to different salinity concentrations in *N. nuchalis* inhabiting the top of the estuary and those inhabiting the deep waters of the outer bay. To perform this, using transcriptomic profiles, we characterized sequence information of the ApoAI gene from *N. nuchalis* inhabiting different coastal environments. We confirmed the environmental niche of *N. nuchalis* using stable isotope analysis to identify the habitat and feeding environment of the fish collected from the top of the estuary (low salinity) and the deep waters (high salinity), because of fish mobility. We then compared the expression pattern of the ApoAI transcript in *N. nuchalis* inhabiting the top of the estuary with those inhabiting the deep-bay areas; the results were correlated with aquatic environmental factors. We aimed to reveal the functional role of ApoAI in different environments of low or high salinity to improve our understanding of the metabolic processes of osmotic adaptation and osmoregulation in *N. nuchalis* living in diverse salinities.

## 2. Materials and Methods

### 2.1. Ethics Statement

The research was conducted in accordance with the guidelines and regulations of the Animal Care and Use Committee of Chonnam National University (Yeosu, South Korea). This study did not involve endangered or protected species.

### 2.2. Sample Collection and Environmental Factor Measurements

Adult *N. nuchalis* were collected in June 2019 from Gwangyang Bay, which is located in the south coast of the Korean peninsula (Figure 1). Using an outboard, fish collection was performed with a dredge net (net width 1.5 × 0.5 m, net length 10 mm). The average body length and weight of the fish were 70.83 ± 10.35 mm and 4.15 ± 1.52 g, respectively (total *n* = 40). In the bay, the water depth varies from 10 m at the Seomjin River estuary to 50 m at the outer bay [3]. The Seomjin River is one of the four major watersheds of South Korea and is located in the southwestern section of the Korean peninsula (34°55′–35°45′ N, 126°57′–127°55′ E). Salinity and water temperature were measured at the collection locations using a portable probe (Professional Plus, YSI, Yellow Springs, OH, USA) (Figure 1 and Table 1). At the sample locations, environmental data and fish were collected and measured using the standard methods described in Kim et al. [3]. Data collected included chlorophyll *a* concentrations as well as organic and inorganic carbon concentrations (Table 1).

### 2.3. Stable Isotope Analysis of the Bulk Fish Tissue

Carbon and nitrogen isotope ratios (δ^13^C and δ^15^N) were analyzed in the *N. nuchalis* samples (*n* = 3 at each ES1 and ES4 site). The pre-treatment methodology described in Choi et al. [27] was followed for inorganic carbon and lipid removal before the bulk carbon isotope analysis. Briefly, each homogenized sample was sequentially treated with 1 M HCl and a mixture of chloroform and methanol (2:1, *v/v*). For the δ^15^N analysis, the homogenized samples were used immediately without any further treatment. δ^13^C and δ^15^N were measured using an elemental analyzer (EA-3000, Eurovector, Pavia, Italy) with an isotope ratio mass spectrometer (Isoprime100, Isoprime Ltd., Stockport, UK). These ratios are expressed as per mil (‰), which is defined as the δ value of the isotope ratio difference between the standard material and the sample. The formula is as follows:*δX* (‰) = [(*R_sample_*/*R_standard_* − 1) × 1000](1)
where *X* is ^13^C or ^15^N and R is the ^13^C/^12^C or ^15^N/^14^N for carbon or nitrogen, respectively. The standards for each carbon and nitrogen value used were Pee Dee Belemnite and air (N_2_), respectively. The degree of analytical precision was within 0.2‰ for both δ^13^C and δ^15^N. The average stable isotope ratios from each species were used for the assessment of the food web structure.

### 2.4. RNA Extraction and Quantitative Real-Time PCR (qRT-PCR)

RNA extraction was performed on *N. nuchalis* collected from five sampling sites (*n* = 3 at each ES1, ES2, ES3, ES4, and ES5 site) within the bay (Figure 1). The total RNA was extracted from the gill tissue of three fish from each sample site using TRIzol (Invitrogen, Carlsbad, CA, USA) and following the manufacturer’s instructions. Purification of the extracted RNA was performed using a RNase-free DNase set (Qiagen, Hilden, Germany) and RNeasy Plus Mini Kit (Qiagen). The concentration and quality of the total RNA samples were determined using 1% agarose gel electrophoresis and a NanoPhotometer spectrophotometer (IMPLEN, CA, USA). The cDNA was synthesized using 1 µg of RNA and the SuperScript^TM^ III RT Kit (Invitrogen). The synthesized cDNA was diluted 40-fold and kept at −80 °C. The quantitative real-time PCR (RT-qPCR) was performed using Accuprep^®^2x Greenstar qPCR Master Mix (Bioneer, Daejon, Korea) and ExicyclerTM 96 (Bioneer). Glyceraldehyde-3-phosphate dehydrogenase (GAPDH) is an internal control to normalization of the relative ApoAI expressional levels. The primer sequences for the RT-qPCR were: ApoAI forward, 5′-AAAGATCTGGCTTCCCCC TA-3′; ApoAI reverse, 5′-TCGAAGATGGTCTGGAGCTT-3′; GAPDH forward, 5′-TGTGCAGCAATGAA AGAAG C-3′; GAPDH reverse, 5′-ACCGATTTCGTTGTCGTACC-3′. The PCR product size was 158 bp for the ApoAI gene and 183 bp for the GAPDH gene, which was used as an internal reference. The PCR machine was operated with the following thermal cycling conditions: 95 °C for 3 min, followed by 40 cycles of 95 °C for 30 s, 54 °C for 40 s, and 72 °C for 20 s. The 2^−ΔΔCt^ method was used to analyze the relative expression levels of the target genes.

### 2.5. Data Analysis

All results are presented as the mean and standard error. The data were analyzed using the Statistical Package for the Social Sciences 16.0 KO (SPSS Inc., Chicago, IL, USA). The expression level of the ApoAI transcripts in each sample was normalized to that of GAPDH. Two-way analysis of variance (ANOVA) was used to identify any significant differences between the different salinity environments and the low salinity sites (ES1-t, ES2). The results were considered significant when *p* < 0.05. Using the PAST 3.23 program, a principal component analysis (PCA) was performed with biplot construction to visualize the relationships between expression levels of the ApoAI transcripts and the aquatic environmental factors.

## 3. Results

### 3.1. Analyses of δ^13^C and δ^15^N in Bulk Tissue from N. nuchalis

The δ^15^N ratio was similar for the *N. nuchalis* samples between ES1 and ES4 (14–15‰; Figure 2A). However, δ^13^C indicated the differences between ES1 and ES4 fish samples with regard to diets with limited discrimination among food sources. The δ^13^C ranged from −19 to −20‰ for ES1 and from −17 to −18‰ for ES4 (Figure 2B).

### 3.2. Characterization and Phylogenetic Analysis of the N. nuchalis ApoAI Gene

The sequence data of the *N. nuchalis* ApoAI gene were obtained from the transcriptome de novo sequencing database using the Illumina platform (data not shown). The *N. nuchalis* ApoAI was 852 bp long, including an open reading frame of 284 amino acids (Figure 3A). The alignment of the amino acid sequences of ApoAI in *N. nuchalis* with those in other fish revealed that the protein contains several residue repeats forming a pair of alpha helices (in the pfam domain) including apolipoprotein families (represented as red rectangles in Figure 3A). The amino acid sequence of ApoAI in *N. nuchalis* was 73%, 72%, 70% and 69% homologous with that in *Echeneis naucrates* (Linnaeus, 1758), *Perca fluviatilis* (Linnaeus, 1758), *Etheostoma cragini* (Gilbert, 1885), and *Morone saxatilis* (Walbaum, 1792), respectively. At the nucleotide level, it was 80%, 78%, and 76% homologous with that in *E. naucrates* (XM_029517515), *M. saxatilis* (XM_035680143), and *Epinephelus coioides* (Hamilton, 1822) (JN540027). The phylogenetic analysis placed the *N. nuchalis* ApoAI sequence in the same clade with ApoAI and ApoAI-like homologs from fish (Figure 3B). There was no sister group of *N. nuchalis* ApoAI among the large clades of fish species. Additionally, there was a significant separation between ApoAI from fish and that from mammalian species, as indicated by their positions in different clusters (Figure 3B).

### 3.3. ApoAI Gene Expressions in N. nuchalis Collected from Different Salinity Environments

The ApoAI gene with a significant fold change was identified from a differentially expression gene (DEG) analysis of the transcriptome de novo sequencing data in *N. nuchalis* groups inhabiting a low salinity range (ES1) and a high salinity range (ES4) in Gwangyang bay (data not shown). Furthermore, the relative level of the ApoAI gene in *N. nuchalis* collected from the high salinity environment (ES4-t) was over 40 times higher than that in *N. nuchalis* collected from the low salinity environment (ES1-t) (Figure 4). To validate the expression patterns of the *N. nuchalis* ApoAI gene in the field environment, a qRT-PCR was conducted using the *N. nuchalis* samples collected from four sites across the bay with different salinity ranges (14.9 psu for ES2, 27.3 psu for ES3, 31.22 psu for ES4, and 31.23 psu for ES5; Figure 1 and Figure 4). The expression level of the ApoAI transcripts was significantly high in the *N. nuchalis* inhabiting the high salinity environment with over 31 psu (ES4 and ES5). This result is consistent with that of the DEG analysis of the transcriptome sequencing data.

### 3.4. PCA Correlation between the ApoAI Transcript Expression and Nine Environmental Factors

Potential correlations between the *N. nuchalis* ApoAI gene expression and nine field measured environmental factors (Table 1) were analyzed using PCA (Figure 5). The first two principal components (Principal Component 1 (PC1)) and (Principal Component 2 (PC2)) cumulatively explained 94.8% (65% by PC1 and 29.8% by PC2) of the total variation. The biplots showed the distribution of the variables, including the ApoAI transcript expression and the nine environmental factors. The PCA revealed that the ApoAI transcript expression was positively correlated with salinity, transparency, pH, TIC, and TC at the high salinity locations ES4 and ES5. Chl-a was positively correlated with TOC and DO at the low salinity locations ES1 and ES2 (Figure 5).

## 4. Discussion

The present study provides the first report of different patterns of ApoAI mRNA expression in *N. nuchalis* inhabiting various environmental salinities. Although *N. nuchalis* are the dominant species in shallow inland bays, there have been very rare studies characterizing genes in the species. In this study, the variation in the δ^13^C values of the *N. nuchalis* collected from different sites of the Gwangyang Bay (estuarine to deep-bay areas) is because of the organic matter at each site. These results allow us to infer the characteristics of the feeding habitats of *N. nuchalis*. Therefore, the differences in δ^13^C of *N. nuchalis* suggest that the feeding activity varied in different habitats. The stable isotope analysis supported to identification of *N. nuchalis* inhabiting different salinity habitats for characterizing osmoregulation-related genes in the field fish. The results of the stable isotope analysis also showed that *N. nuchalis* has a broad feeding range from zooplanktonic (estuarine habitat) to epibenthic (deep-bay habitat) during their migration between these habitats of different salinity ranges [9]. The stable isotope analysis identified the habitat characteristics preferred by this transient fish species, allowing gene expression pattern comparisons with field caught individuals.

ApoAI is a main and abundant plasma high-density lipoprotein protein that is known to possess diverse protective functions such as antioxidant, anti-inflammatory, anti-atherosclerotic, and anti-thrombotic activities [19,28]. ApoAI is involved in the innate immune system and has a pivotal role in early defense responses against pathogen infection [23]. ApoAI forms cationic amphipathic α-helices, intercalates into lipid bilayers, neutralizes lipopolysachharides (the major pathogenic factor), and inhibits inflammatory cytokines [21,29]. The antibacterial activity of ApoA1 has been reported in several teleosts, including spotted sea bass (protection from *Vibrio harveyi* infection), channel catfish (protection from *Aeromonas hydrophila* infection), and the orange-spotted grouper (protection from *Microcococcus lysodeikticus* infection) [21,23,24]. An interesting finding of the present study is the differential expression of ApoAI mRNAs in *N. nuchalis* inhabiting different salinities from the top of the estuary to the outer bay. We identified that, at the transcriptional level, the ApoAI gene was significantly higher in *N. nuchalis* inhabiting the high salinity locations (over 31 psu) of the bay than in those inhabiting the low salinity locations (under 10 psu). The differential expression of the ApoAI proteins and mRNA has also been observed in ayu (*Plecoglossus altivelis*, Temminck and Schlegel, 1846) sampled from freshwater and brackish water [18]. These results suggest an essential function of ApoAI in the hyperosmotic regulation within fish. Moreover, changes in ApoAI transcript expression in *N. nuchalis* can reflect potential differences in the defense process (such as innate immune signaling in response to pathogen infection) with varying habitat conditions, such as salinity.

Previously, the distribution pattern of *N. nuchalis* was assumed to differ based on body size, with smaller individuals inhabiting the low-saline estuary and larger fish inhabiting the deep-bay areas, even though the fish is widely distributed throughout the bay area, reflecting its euryhaline characteristics [9]. The habitat of the large *N. nuchalis* fish (over 80 mm) is limited to Gwangyang Bay (over 30 psu). ApoAI also plays a key role in lipid transport in the circulatory system [18,21]. Fish use lipids as their main energy source, which is in contrast to mammals that mainly use carbohydrates. Therefore, lipid metabolism through lipoprotein activities appears to be crucial for homeostasis maintenance in fish [30,31]. In addition, fish ApoAI is required for normal embryonic development, ontogenic growth, and tissue regeneration [32,33]. Based on the findings of this study, expressional changes in the *N. nuchalis* ApoAI gene may be related to alterations in the lipid and glucose metabolism and may therefore induce differential growth and development, resulting in different body sizes of *N. nuchalis* fish. The ApoAI mRNA levels are more closely associated with larval size and development than dietary lipid levels in Senegalese sole (*Solea senegalensis,* Kaup, 1858) [34].

## 5. Conclusions

In this study, we provide transcriptional responses of ApoAI gene in *N. nuchalis* inhabiting different salinity environments from the estuary to the deep water using transcriptomic profiles and gene expression analysis. For characterizing osmoregulation-related genes in field fish, stable isotope analysis was first applied to identify habitat and feeding sources of *N. nuchalis* collected from the estuary water with a low salinity to the deep ocean water with a high salinity level. The ApoAI expression pattern correlated with environmental parameters suggest the possibility as a molecular biomarker for monitoring effects of salinity variation in fish. Consequently, the present study suggested that ApoAI gene expression can apply to assessing osmotic defense function and habitat characteristics of *N. nuchalis* traversing broad salinity ranges, and will help to further understand functional process of osmoregulation and lipid metabolism in fish. Furthermore, this study for gene characterization in fish will be useful in the identification of bioactive molecules for industrial applications using fish [35] as well as fish health and environmental monitoring.

## Figures and Tables

**Figure 1 ijerph-18-10960-f001:**
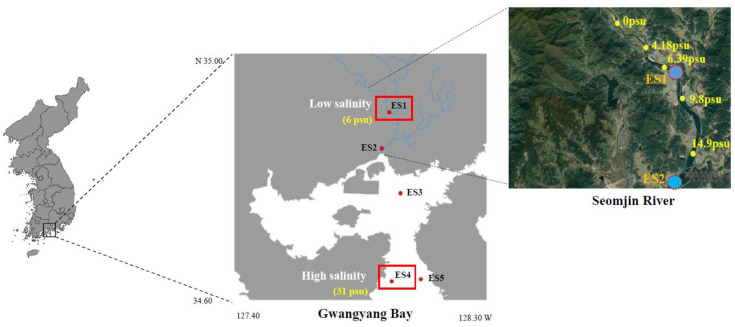
Map of the collection location sites for *N. nuchalis* from the estuary to the deep-bay areas of Gwangyang bay. The red rectangles indicated sampling sites for the transcriptome de novo sequencing analysis.

**Figure 2 ijerph-18-10960-f002:**
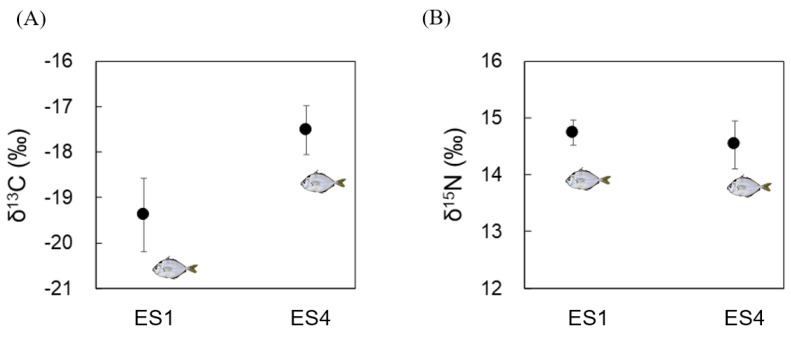
Carbon and nitrogen isotope ratios in *N. nuchalis* sampled from the Seomjin estuary (ES1: low salinity) and Gwanyang Bay (ES4: high salinity). (**A**) Carbon stable isotope ratios (δ^13^C); (**B**) nitrogen stable isotope ratios (δ^15^N).

**Figure 3 ijerph-18-10960-f003:**
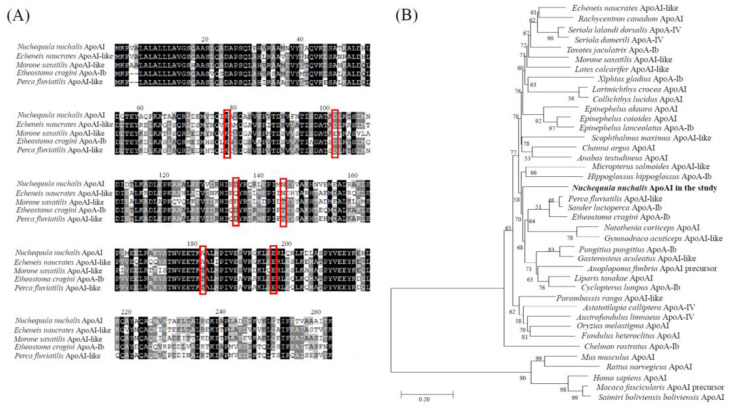
Characterization of the *N. nuchalis* ApoAI gene. (**A**) Multiple-sequence alignment of the *N. nuchalis* ApoAI gene sequences with the homologous sequences of other fish species. The red rectangles represent residue repeats that form a pair of alpha helices including apolipoprotein families. (**B**) Phylogenetic relationship of the *N. nuchalis* ApoAI gene with other reported ApoAIs. The phylogenetic tree of the aligned amino acid sequences was constructed by a neighbor-joining analysis using the MEGA 4.0 program. Bootstrap values (1000 replicates) are indicated at the nodes. The bar indicates the genetic distances (0.02). The species information was presented in in Appendix A.

**Figure 4 ijerph-18-10960-f004:**
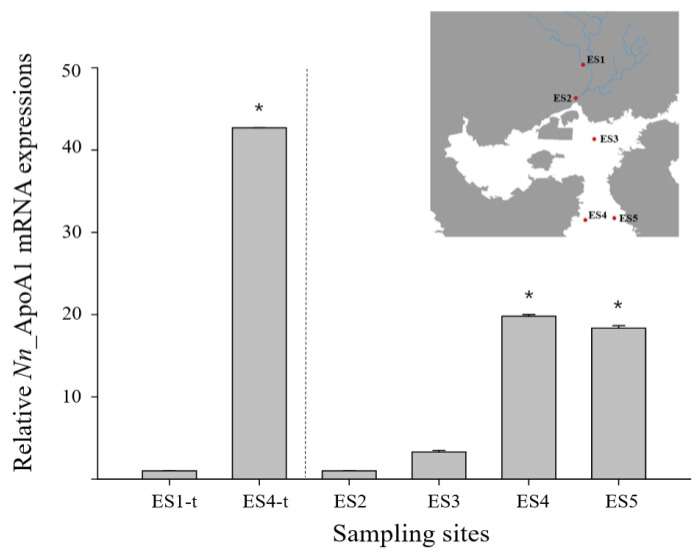
Relative expression of *N. nuchalis* ApoAI mRNA in *N. nuchalis* collected from field environments with different salinity ranges. ES1-t and ES4-t present the expression results of the *N. nuchalis* ApoAI transcript analysis using the transcriptome de novo sequencing database. The values were normalized against those of GAPDH. Values of each bar represent the mean ± SD. A statistically significant difference (*p* < 0.05) is presented by an asterisk and compares the results with those of the low salinity samples (relative value of ApoAI in ES1-t or ES2 = 1).

**Figure 5 ijerph-18-10960-f005:**
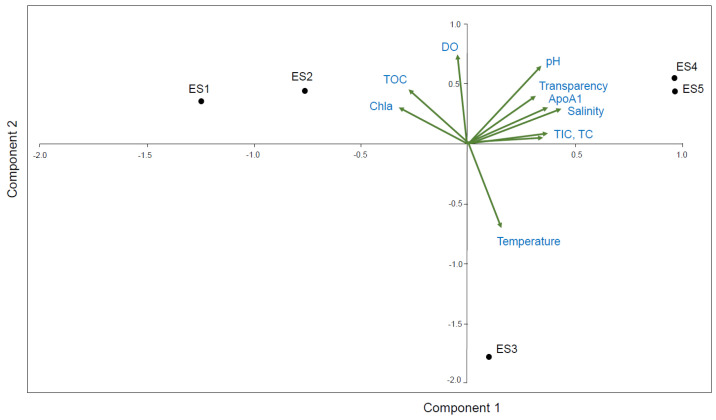
Principal component analysis (PCA) for correlation between the *N. nuchalis* ApoAI expression patterns and nine environmental factors (salinity, temperature, transparency, DO, pH, TOC, TIC, TC, and Chl-a). Relationships between ApoAI gene expression and environmental factors and fish sampling sites (ES1, ES2, ES3, ES4, and ES5) with different salinity ranges in PC1 and PC2.

**Table 1 ijerph-18-10960-t001:** Water quality parameters of the five collection locations in Gwangyang bay.

	Salinity (psu)	Temperature (°C)	Transparency (cm)	DO (mg L^−1^)	pH	TOC (mg L^−1^)	TIC (mg L^−1^)	TC (mg L^−1^)	Chl-a (μg L^−1^)
ES1	6.39	20	135	7.99	7.6	3.617	7.546	11.163	4.93
ES2	14.9	20.6	175	7.8	7.83	3.961	9.32	13.281	4.64
ES3	27.3	22.7	200	6.68	7.43	2.562	10.612	13.174	1.3
ES4	31.22	21.3	360	8.05	8.19	2.779	12.572	15.35	0.87
ES5	31.23	20.7	350	7.66	8.14	2.932	14.084	17.015	2.09

Dissolved oxygen (DO), total organic carbon (TOC), total inorganic carbon (TIC), total carbon (TC), and chlorophyll-a (Chl-a).

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
