# Peer review of "Differential Expression of the Apolipoprotein AI Gene in Spotnape Ponyfish (Nuchequula nuchalis) Inhabiting Different Salinity Ranges at the Top of the Estuary and in the Deep-Bay Area of Gwangyang Bay, South Korea"

_ijerph, 2021, doi:10.3390/ijerph182010960_

Round 1

Reviewer 1 Report

Review comments on the manuscript by Park et al

Differential expression of the aliopoprotein AI gene in spotnape ponyfish (Leiognathus nuchalis) inhabiting different salinity ranges at the top of the estuary and the deep-bay area of Gwangyang Bay, South Korea

The aim of the study was to characterize the Alipoprotein AI gene in the ponyfish Leiognathus nuchalis and to analyse its expression in fish from a range of salinity habitats in a coastal bay in South Korea. The habitat characteristics and diet of fish from different areas were confirmed by stable isotope analysis. The study found that the transcriptional expression level of ApoAI was higher in fish occurring in more saline habitats compared with lower salinity habitats further up the estuaries.

Overall, the data presented achieves the study aims and provides an original contribution to our knowledge of the transcriptional expression level of ApoAI in fish and its role in fish physiology in relation to osmoregulation. The analytical techniques and QA/QC procedures, sampling design and statistical analysis are appropriate to address the aims of the study and the tables and figures relevant to presenting the results of the study. As such, I consider that a revised paper incorporating the marked comments on the attached PDF and addressing the comments detailed below could be resubmitted for assessment for publication.

In Figure 1, the site ES1 is shown, but it is not included in Table 1. In addition, no reference is made to sites ES4 or ES6 in Figure 1 or Table 1. This indicates that the sampling sites for this study were sampled in a separate study, but no detail is provided in this paper to clarify this issue. Alternatively, the sites for this paper should be re-numbered from S1 to S5 to remove confusion.

Section 2.3 implies that bulk fish tissue for isotope analysis occurred for fish from the five sampling sites. However, in section 3.1, isotope data is only presented in for sites ES2 and ES7 with no isotope data provided for the other sites. Was there data for these three sites? If not, the methods section should be updated to clarify this.

In line 165 for site ES2 a salinity data range of 1.57- 18.51 psu is reported while in table 1 only a value of 6.39 psu is reported.  Similarly for ES7, a range of 31.22-32.01 psu is reported versus a value of 31.22 in Table 1.  The authors should clarify these differences.

Given that the author’s first language is not English, I have marked on the attached PDF some suggested spelling and grammatical changes to improve the quality and flow of the manuscript. The authors should review these changes and have the manuscript reviewed before resubmission.

There are also a number of issues with the references. The authors should check previous issues of the journal regarding presentation of references.

Author Response

4st October 2021

Editors

Dear Editors:

I am sending the revised manuscript titled “Differential expression of the apolipoprotein AI gene in spotnape ponyfish (Nuchequula nuchalis) inhabiting different salinity ranges at the top of the estuary and in the deep-bay area of Gwangyang Bay, South Korea” for consideration to publish in your prestige journal.

Spotnape ponyfish (Nuchequula nuchalis) is a dominant species that is broadly distributed from estuarine to deep-bay areas, reflecting a euryhaline habitat. Apolipoprotein AI (ApoAI) is a main component of plasma lipoproteins and has crucial roles in lipid metabolism and the defense immune system. In this study, we characterized the N. nuchalis ApoAI gene and analyzed the expression of the ApoAI transcript in N. nuchalis collected at various sites in the estuary and the deep-bay area which have different salinities. Owing to the fish’s mobility, we conducted stable isotope analyses to confirm the habitat characteristics of N. nuchalis. Carbon and nitrogen isotope ratios (δ13C and δ15N) from N. nuchalis indicated different feeding sources and trophic levels in the estuarine and deep-bay habitats. The characterized N. nuchalis ApoAI displayed residual repeats that formed a pair of alpha helices, indicating that the protein belongs to the apolipoprotein family. In the phylogenetic analysis, there was no sister group of N. nuchalis ApoAI among the large clades of fish species. The transcriptional expression level of ApoAI was higher in N. nuchalis inhabiting the deep-bay area with a high salinity (over 31 psu) than in N. nuchalis inhabiting the top of the estuary with a low salinity (6~15 psu). In addition, the expression patterns of N. nuchalis ApoAI were positively correlated with environmental factors (transparency, pH, TC, and TIC) in the high salinity area. These results suggest that ApoAI gene expression can reflect habitat characteristics of N. nuchalis which traverse broad salinity ranges and is associated with functional roles of osmoregulation and lipid metabolism for fish growth and development

We believe that our findings would satisfy your journal requirements for the publication of this paper. Therefore, please be kind enough to evaluate our manuscript and inform us your comments. The English in the manuscript has been corrected by a professional science-editing service (Editage). In addition, our specific responses to the points raised by the reviewers are presented below.

Reviewer Comments:

Reviewer #1

Comments and Suggestions for Authors

Review comments on the manuscript by Park et al

Differential expression of the aliopoprotein AI gene in spotnape ponyfish (Leiognathus nuchalis) inhabiting different salinity ranges at the top of the estuary and the deep-bay area of Gwangyang Bay, South Korea

The aim of the study was to characterize the Alipoprotein AI gene in the ponyfish Leiognathus nuchalis and to analyse its expression in fish from a range of salinity habitats in a coastal bay in South Korea. The habitat characteristics and diet of fish from different areas were confirmed by stable isotope analysis. The study found that the transcriptional expression level of ApoAI was higher in fish occurring in more saline habitats compared with lower salinity habitats further up the estuaries.

Overall, the data presented achieves the study aims and provides an original contribution to our knowledge of the transcriptional expression level of ApoAI in fish and its role in fish physiology in relation to osmoregulation. The analytical techniques and QA/QC procedures, sampling design and statistical analysis are appropriate to address the aims of the study and the tables and figures relevant to presenting the results of the study. As such, I consider that a revised paper incorporating the marked comments on the attached PDF and addressing the comments detailed below could be resubmitted for assessment for publication.

In Figure 1, the site ES1 is shown, but it is not included in Table 1. In addition, no reference is made to sites ES4 or ES6 in Figure 1 or Table 1. This indicates that the sampling sites for this study were sampled in a separate study, but no detail is provided in this paper to clarify this issue. Alternatively, the sites for this paper should be re-numbered from S1 to S5 to remove confusion.

-->We revised in the all Figures, Table 1 and revised manuscript with re-numbered sampling sites (from ES1 to ES5) as the reviewer’s comment.

Section 2.3 implies that bulk fish tissue for isotope analysis occurred for fish from the five sampling sites. However, in section 3.1, isotope data is only presented in for sites ES2 and ES7 with no isotope data provided for the other sites. Was there data for these three sites? If not, the methods section should be updated to clarify this.

-->We conducted stable isotope analyses to confirm the habitat characteristics of L. nuchalis at low salinity site (ES1) and high salinity site (ES4) for transcriptomic analysis to characterize osmoregulation related gene such as the ApoA1. In line 125, we revised from “at each site” to “at ES1 and ES4 sites” in the revised manuscript.

In line 165 for site ES2 a salinity data range of 1.57- 18.51 psu is reported while in table 1 only a value of 6.39 psu is reported.  Similarly for ES7, a range of 31.22-32.01 psu is reported versus a value of 31.22 in Table 1.  The authors should clarify these differences.

-->we revised and moved the sentence to lines 81-84 of the introduction part in the revised manuscript.

Given that the author’s first language is not English, I have marked on the attached PDF some suggested spelling and grammatical changes to improve the quality and flow of the manuscript. The authors should review these changes and have the manuscript reviewed before resubmission.

-->We revised all spelling and grammatical errors as the reviewer’s comments on the attached PDF in the revised manuscript. We marked the revised points as the shaded yellow in the revised manuscript.

There are also a number of issues with the references. The authors should check previous issues of the journal regarding presentation of references.

-->We checked and revised all references format in the revised manuscript.

Thanks in advance.

Yours sincerely,

Ihn-Sil Kwak, Ph.D.

Dept. of Ocean Integrated Science,

Chonnam National University, Chonnam, 550-749, Korea.

E-mail address: iskwak@chonnam.ac.kr/inkwak@hotmail.com

Reviewer 2 Report

GENERAL COMMENTS.

Present MS about differential expression of the apolipoprotein AI gene in Leiognathus nuchalis from different areas with different salinity range is interesting and of value to deepen the knowledge on the physiology and adaptation of this species to euryhaline habitats. Increasing the knowledges about fishes’ biological response to environmental variations is essential to understand their life cycles and ecology, to make conservation more effective.

However, I do have some major concerns that I would like to see resolved before the publication.

Some manuscript chapters need a better organization, and, on my opinion, there are some references lacks. The results and the conclusion are not concisely written and are presented in a confusing way making it for readers extremely difficult to get the main point of the study.

Although the text is mostly easy to understand, there are quite some grammatical errors. I recommend an English proofreading.

SPECIFIC COMMENTS

Title:

The species name of studied species is not presented in the correct way, missing species authorities: Leiognathus nuchalis(Temminck & Schlegel, 1845). Moreover, this name is unaccepted. The accepted name for the species is Nuchequula nuchalis (Temminck & Schlegel, 1845). Please reword the species name in the entire manuscript and make attention to the scientific name of other species (e.g., line 46, Argyrosomus argentatus (Houttuyn, 1782) is unaccepted; the accepted name is Pennahia argentata (Houttuyn, 1782). See WORMS (http://www.marinespecies.org/index.php), the world register of marine species, to see all the species authorities and accepted name.

I suggest, the first way in which a species is cited within the text, it should be write also species authorities, e.g., line 192 (Perca fluviatilis, Linnaeus, 1758). Please make more attention to scientific name of all the species cited in MS.

Introduction:

Line 45: Please cite the references regarding Gwangyang Bay species richness and diversity in 2018.

Line 50: It should be added the ecological information about studied species (e.g., trophic position, feeding habit, diet variations)

Line 65: I strongly suggest adding references regarding other biological molecules and genes involved in antimicrobial activities and innate immunity, to expand the introduction:

Abbate, J.M.; Macrì, F.; Capparucci, F.; Iaria, C.; Briguglio, G.; Cicero, L.; Salvo, A.; Arfuso, F.; Ieni, A.; Piccione, G.; Lanteri, G. Administration of Protein Hydrolysates from Anchovy (Engraulis Encrasicolus) Waste for Twelve Weeks Decreases Metabolic Dysfunction-Associated Fatty Liver Disease Severity in ApoE–/–Mice. Animals 2020, 10, 2303. https://doi.org/10.3390/ani10122303 ;

Abbate, J.M.; Macrì, F.; Arfuso, F.; Iaria, C.; Capparucci, F.; Anfuso, C.; Ieni, A.; Cicero, L.; Briguglio, G.; Lanteri, G. Anti-Atherogenic Effect of 10% Supplementation of Anchovy (Engraulis encrasicolus) Waste Protein Hydrolysates in ApoE-Deficient Mice. Nutrients 2021, 13, 2137. https://doi.org/10.3390/nu13072137 ;

Capillo, G., Zaccone, G., Cupello, C., Fernandes, J. M. O., Viswanath, K., Kuciel, M., ... & Lauriano, E. R. (2021). Expression of acetylcholine, its contribution to regulation of immune function and O2 sensing and phylogenetic interpretations of the African butterfly fish Pantodon buchholzi (Osteoglossiformes, Pantodontidae). Fish & Shellfish Immunology111, 189-200.

Lauriano, E. R., Pergolizzi, S., Aragona, M., Montalbano, G., Guerrera, M. C., Crupi, R., ... & Capillo, G. (2019). Intestinal immunity of dogfish Scyliorhinus canicula spiral valve: a histochemical, immunohistochemical and confocal study. Fish & shellfish immunology87, 490-498.

Line 72:  On my opinion it is not clear the reason why stable isotope analysis is required for present MS and what they add to the investigation about the gene expression. Please, develop this point.

Materials and Methods:

Line 85: How were the individual collected? With what kind of fishing? Please add this information!

Line 86: How many individuals were collected and analyzed for present MS?

Line 114 – 116: These sentences should be moved in Introduction. Please, make attention! In Materials and Methods chapter, the authors should only present the applied methodology and the materials used for MS.

Results:

Line 163 – 166: This sentence should not be in Results chapter. It is an introductory sentence. Please make attention! In Results chapter the author should only present results, without comment or introductory. I strongly recommend to re-structure the results paragraphs, making attention to the introductory sentences.

Line 166: “an indicator of trophic positions” is an unnecessary sentence; it has already been specified in Materials and Methods.

Line 244 – 247: The sentence should not be in Results chapter. See the comment above (Line 163 – 166).

Figure captions:

Figure 1: Why ES2 and ES7 are with a box border? Please add this information in figure caption.

Figure 3: The caption is written in a confusing way, and it is hard to read. I strongly recommend deleting the GenBank accession numbers for the several species.

Figure 4: The caption is written on a confusing way, please reword it. Moreover, what is GADPH? Please explain it in caption or in MS chapters.

Discussion:

Line 305 – 308: This sentence is unappropriated and redundant; L. nuchalis biological and ecological feature have been explained in Introduction. Please add this information in that MS part.

Line 318 – 320: I still don’t understand the reason why stable isotope analysis is required for present MS and what they add to the investigation about the gene expression. The authors should develop this part of discussion explaining why this analysis is of value for present MS.

Line 325: The authors should fix ApoA1 with ApoAI.

Line 354: Pleas add to Senegalese solea the accepted scientific name with species authorities. I strongly recommend doing this in the entire MS every first way in which a species is cited (See Title comments).

Conclusion:

I strongly recommend to thoroughly re-structure the Conclusion. The authors should expand them, deepen the prospects open by present MS and what novelty has been added in this research field.

For example, an aspect that could be deepened by authors should be the fish waste reusing for bioactive molecules extraction. See and cite:

Gervasi, T., Santini, A., Daliu, P., Salem, A. Z., Gervasi, C., Pellizzeri, V., ... & Cicero, N. (2020). Astaxanthin production by Xanthophyllomyces dendrorhous growing on a low-cost substrate. Agroforestry Systems, 94(4), 1229-1234.

Gervasi, T., Pellizzeri, V., Benameur, Q., Gervasi, C., Santini, A., Cicero, N., & Dugo, G. (2018). Valorization of raw materials from agricultural industry for astaxanthin and β-carotene production by Xanthophyllomyces dendrorhous. Natural product research, 32(13), 1554-1561 ;

Line 357 – 367: These sentences should be deleted; they are redundant and unappropriated for Conclusion.

Author Response

4st October 2021

Editors

Dear Editors:

I am sending the revised manuscript titled “Differential expression of the apolipoprotein AI gene in spotnape ponyfish (Nuchequula nuchalis) inhabiting different salinity ranges at the top of the estuary and in the deep-bay area of Gwangyang Bay, South Korea” for consideration to publish in your prestige journal.

Spotnape ponyfish (Nuchequula nuchalis) is a dominant species that is broadly distributed from estuarine to deep-bay areas, reflecting a euryhaline habitat. Apolipoprotein AI (ApoAI) is a main component of plasma lipoproteins and has crucial roles in lipid metabolism and the defense immune system. In this study, we characterized the N. nuchalis ApoAI gene and analyzed the expression of the ApoAI transcript in N. nuchalis collected at various sites in the estuary and the deep-bay area which have different salinities. Owing to the fish’s mobility, we conducted stable isotope analyses to confirm the habitat characteristics of N. nuchalis. Carbon and nitrogen isotope ratios (δ13C and δ15N) from N. nuchalis indicated different feeding sources and trophic levels in the estuarine and deep-bay habitats. The characterized N. nuchalis ApoAI displayed residual repeats that formed a pair of alpha helices, indicating that the protein belongs to the apolipoprotein family. In the phylogenetic analysis, there was no sister group of N. nuchalis ApoAI among the large clades of fish species. The transcriptional expression level of ApoAI was higher in N. nuchalis inhabiting the deep-bay area with a high salinity (over 31 psu) than in N. nuchalis inhabiting the top of the estuary with a low salinity (6~15 psu). In addition, the expression patterns of N. nuchalis ApoAI were positively correlated with environmental factors (transparency, pH, TC, and TIC) in the high salinity area. These results suggest that ApoAI gene expression can reflect habitat characteristics of N. nuchalis which traverse broad salinity ranges and is associated with functional roles of osmoregulation and lipid metabolism for fish growth and development

We believe that our findings would satisfy your journal requirements for the publication of this paper. Therefore, please be kind enough to evaluate our manuscript and inform us your comments. The English in the manuscript has been corrected by a professional science-editing service (Editage). In addition, our specific responses to the points raised by the reviewers are presented below.

Reviewer #2

Comments and Suggestions for Authors

GENERAL COMMENTS.

Present MS about differential expression of the apolipoprotein AI gene in Leiognathus nuchalis from different areas with different salinity range is interesting and of value to deepen the knowledge on the physiology and adaptation of this species to euryhaline habitats. Increasing the knowledges about fishes’ biological response to environmental variations is essential to understand their life cycles and ecology, to make conservation more effective.

However, I do have some major concerns that I would like to see resolved before the publication.

Some manuscript chapters need a better organization, and, on my opinion, there are some references lacks. The results and the conclusion are not concisely written and are presented in a confusing way making it for readers extremely difficult to get the main point of the study.

Although the text is mostly easy to understand, there are quite some grammatical errors. I recommend an English proofreading.

SPECIFIC COMMENTS

Title:

The species name of studied species is not presented in the correct way, missing species authorities: Leiognathus nuchalis(Temminck & Schlegel, 1845). Moreover, this name is unaccepted. The accepted name for the species is Nuchequula nuchalis (Temminck & Schlegel, 1845). Please reword the species name in the entire manuscript and make attention to the scientific name of other species (e.g., line 46, Argyrosomus argentatus (Houttuyn, 1782) is unaccepted; the accepted name is Pennahia argentata (Houttuyn, 1782). See WORMS (http://www.marinespecies.org/index.php), the world register of marine species, to see all the species authorities and accepted name.

I suggest, the first way in which a species is cited within the text, it should be write also species authorities, e.g., line 192 (Perca fluviatilis, Linnaeus, 1758). Please make more attention to scientific name of all the species cited in MS.

-->We checked all the species authorities and accepted scientific name on the search FishBase (www.fishbase.se) and WORMS (http://www.marinespecies.org/index.php). As the reviewer’s suggestion, we added species authorities such as Perca fluviatilis (Linnaeus, 1758) in the lines 51-52, lines 190-194, and line 208 (in the Supplementary Table 1) of the revised manuscript, marked by yellow shade box.

Introduction:

Line 45: Please cite the references regarding Gwangyang Bay species richness and diversity in 2018.

-->We added the reference ([5]; Kim et al., 2019) regarding Gwangyang Bay species richness and diversity in 2018.

Kim, D.K.; Jo, H.; Han, I.; Kwak, I.S. Explicit Characterization of Spatial Heterogeneity Based on Water Quality, Sediment Contamination, and Ichthyofauna in a Riverine-to-Coastal Zone. Int. J. Environ. Res. Public Health 2019, 16, 409.

Line 50: It should be added the ecological information about studied species (e.g., trophic position, feeding habit, diet variations)

-->We added the ecological information of N. nuchalis in the lines 52-60 in the revised manuscript.

Line 65: I strongly suggest adding references regarding other biological molecules and genes involved in antimicrobial activities and innate immunity, to expand the introduction:

-->We added the sentence and references regarding other biological molecules and genes involved in antimicrobial activities and innate immunity in lines 74-76.

Abbate, J.M.; Macrì, F.; Capparucci, F.; Iaria, C.; Briguglio, G.; Cicero, L.; Salvo, A.; Arfuso, F.; Ieni, A.; Piccione, G.; Lanteri, G. Administration of Protein Hydrolysates from Anchovy (Engraulis Encrasicolus) Waste for Twelve Weeks Decreases Metabolic Dysfunction-Associated Fatty Liver Disease Severity in ApoE–/–Mice. Animals 2020, 10, 2303. https://doi.org/10.3390/ani10122303 ;

Abbate, J.M.; Macrì, F.; Arfuso, F.; Iaria, C.; Capparucci, F.; Anfuso, C.; Ieni, A.; Cicero, L.; Briguglio, G.; Lanteri, G. Anti-Atherogenic Effect of 10% Supplementation of Anchovy (Engraulis encrasicolus) Waste Protein Hydrolysates in ApoE-Deficient Mice. Nutrients 2021, 13, 2137. https://doi.org/10.3390/nu13072137 ;

Capillo, G., Zaccone, G., Cupello, C., Fernandes, J. M. O., Viswanath, K., Kuciel, M., ... & Lauriano, E. R. (2021). Expression of acetylcholine, its contribution to regulation of immune function and O2 sensing and phylogenetic interpretations of the African butterfly fish Pantodon buchholzi (Osteoglossiformes, Pantodontidae). Fish & Shellfish Immunology111, 189-200.

Lauriano, E. R., Pergolizzi, S., Aragona, M., Montalbano, G., Guerrera, M. C., Crupi, R., ... & Capillo, G. (2019). Intestinal immunity of dogfish Scyliorhinus canicula spiral valve: a histochemical, immunohistochemical and confocal study. Fish & shellfish immunology87, 490-498.

Line 72:  On my opinion it is not clear the reason why stable isotope analysis is required for present MS and what they add to the investigation about the gene expression. Please, develop this point.

-->The goal of our study is to find candidate genes (suah as ApoAI) with different transcriptional expression in fish living in regions (one is low salinity area from the top of estuary and another is high salinity area from the deep water) with different salinity gradients and apply them to the field environmental monitoring. So, we identified the habitat and food sources of N. nuchalis samples through stable isotope analysis to determine whether the fish lived in low-salinity freshwater and high-salinity oceans. In fish living in different salinity ranges identified by stable isotope analysis, differentially expression genes (DEGs) genes were characterized through transcriptome analysis. The ApoAI gene is one of the characterized genes from transcriptomic results.
In the lines 81-84, we added the reason why stable isotope analysis is required for analysis of ApoAI expression pattern in N. nuchalis inhabiting from the top of the estuary (low salinity) and the deep waters (high salinity).

Materials and Methods:

Line 85: How were the individual collected? With what kind of fishing? Please add this information!

-->We added collected individuals and kind of fishing in lines 99-100.

Line 86: How many individuals were collected and analyzed for present MS?

-->We added individuals collected or analyzed for present MS in lines 101-102, 124-125, and 141-142 in the revised manuscript.

Line 114 – 116: These sentences should be moved in Introduction. Please, make attention! In Materials and Methods chapter, the authors should only present the applied methodology and the materials used for MS.

-->We revised the sentence (lines 114-116 of the original MS) in the introduction part (lines 82-84 of the revised manuscript).

Results:

Line 163 – 166: This sentence should not be in Results chapter. It is an introductory sentence. Please make attention! In Results chapter the author should only present results, without comment or introductory. I strongly recommend to re-structure the results paragraphs, making attention to the introductory sentences.

-->We moved the sentence to the introduction part (lines 82-84) of the revised manuscript.

Line 166: “an indicator of trophic positions” is an unnecessary sentence; it has already been specified in Materials and Methods.

-->We deleted the sentence “an indicator of trophic positions” in line 174 of the revised manuscript.

Line 244 – 247: The sentence should not be in Results chapter. See the comment above (Line 163 – 166).

--> In lines 211-214, these sentences are presented to results of transcriptome de novo sequencing data in N. nuchalis groups inhabiting a low salinity range and a high salinity range in Gwangyang bay, to identify ApoAI gene as a differentially expression gene (DEG). The transcriptomic results indicated the relative expression levels of ES1-t and ES4-t on the Fig. 4.

Figure captions:

Figure 1: Why ES2 and ES7 are with a box border? Please add this information in figure caption.

-->We added the information of a box border in Fig. 1 caption.

Figure 3: The caption is written in a confusing way, and it is hard to read. I strongly recommend deleting the GenBank accession numbers for the several species.

-->We revised Fig. 3 caption and species information including the Genbank accession Num. summarized in the new supplementary Table 1.

Figure 4: The caption is written on a confusing way, please reword it. Moreover, what is GADPH? Please explain it in caption or in MS chapters.

-->Lines 152-153, we added the information of GAPDH in the material and method section. Glyceraldehyde-3-phosphate dehydrogenase (GAPDH) is an internal control to normalization of the relative ApoAI expressional levels.

Discussion:

Line 305 – 308: This sentence is unappropriated and redundant; L. nuchalis biological and ecological feature have been explained in Introduction. Please add this information in that MS part.

-->Lines 253-255, we revised and deleted the commented sentences in the discussion part.

Line 318 – 320: I still don’t understand the reason why stable isotope analysis is required for present MS and what they add to the investigation about the gene expression. The authors should develop this part of discussion explaining why this analysis is of value for present MS.

-->As I mentioned earlier, the goal of our study is to find candidate genes (suah as ApoAI) with different transcriptional expression in fish living in regions (one is low salinity area from the top of estuary and another is high salinity area from the deep water) with different salinity gradients and apply them to the field environmental monitoring. So, we identified the habitat and food sources of N. nuchalis samples through stable isotope analysis to determine whether the fish lived in low-salinity freshwater and high-salinity oceans.In fish living in different salinity ranges identified by stable isotope analysis, differentially expression genes (DEGs) genes were characterized through transcriptome analysis. The ApoAI gene is one of the characterized genes from transcriptomic results.

In lines 258-261, we added the meaning for stable isotope data in the present MS.

Line 325: The authors should fix ApoA1 with ApoAI.

-->We revised the word (from ApoA1 to ApoAI) in line 271 of the revised manuscript.

Line 354: Pleas add to Senegalese solea the accepted scientific name with species authorities. I strongly recommend doing this in the entire MS every first way in which a species is cited (See Title comments).

-->We revised as “Senegalese sole (Solea senegalensis, Kaup, 1858)” in line 301 of the revised manuscript.

Conclusion:

I strongly recommend to thoroughly re-structure the Conclusion. The authors should expand them, deepen the prospects open by present MS and what novelty has been added in this research field.

For example, an aspect that could be deepened by authors should be the fish waste reusing for bioactive molecules extraction. See and cite:

-->In lines 303-316, we rewrite the whole conclusion part in the revised manuscript as commented by reviewer.

Gervasi, T., Santini, A., Daliu, P., Salem, A. Z., Gervasi, C., Pellizzeri, V., ... & Cicero, N. (2020). Astaxanthin production by Xanthophyllomyces dendrorhous growing on a low-cost substrate. Agroforestry Systems, 94(4), 1229-1234.

Gervasi, T., Pellizzeri, V., Benameur, Q., Gervasi, C., Santini, A., Cicero, N., & Dugo, G. (2018). Valorization of raw materials from agricultural industry for astaxanthin and β-carotene production by Xanthophyllomyces dendrorhous. Natural product research, 32(13), 1554-1561;

Line 357 – 367: These sentences should be deleted; they are redundant and unappropriated for Conclusion.

-->In lines 303-316, we deleted and revised the whole conclusion part in the revised manuscript.

Thanks in advance.

Yours sincerely,

Ihn-Sil Kwak, Ph.D.

Dept. of Ocean Integrated Science,

Chonnam National University, Chonnam, 550-749, Korea.

E-mail address: iskwak@chonnam.ac.kr/inkwak@hotmail.com

Round 2

Reviewer 2 Report

Dear Authors,

Thanks for addressing most of my comments. I found once again errors in some species name. Please be sure to write, also in the title the correct species name, according to zoological code; for example Nuchequula nuchalis (Temminck & Schlegel, 1845). From the second time you can use the shortened form: N. nuchalis. For example at lines 75-76 Engraulis encrasicolus is written wrongly. 

Please check throughout the entire manuscript.

All the best regards

The Reviewer 

Author Response

8st October 2021

Editors

Dear Editors:

I am sending the revised manuscript titled “Differential expression of the apolipoprotein AI gene in spotnape ponyfish (Nuchequula nuchalis) inhabiting different salinity ranges at the top of the estuary and in the deep-bay area of Gwangyang Bay, South Korea” for consideration to publish in your prestige journal.

Spotnape ponyfish (Nuchequula nuchalis) is a dominant species that is broadly distributed from estuarine to deep-bay areas, reflecting a euryhaline habitat. Apolipoprotein AI (ApoAI) is a main component of plasma lipoproteins and has crucial roles in lipid metabolism and the defense immune system. In this study, we characterized the N. nuchalis ApoAI gene and analyzed the expression of the ApoAI transcript in N. nuchalis collected at various sites in the estuary and the deep-bay area which have different salinities. Owing to the fish’s mobility, we conducted stable isotope analyses to confirm the habitat characteristics of N. nuchalis. Carbon and nitrogen isotope ratios (δ13C and δ15N) from N. nuchalis indicated different feeding sources and trophic levels in the estuarine and deep-bay habitats. The characterized N. nuchalis ApoAI displayed residual repeats that formed a pair of alpha helices, indicating that the protein belongs to the apolipoprotein family. In the phylogenetic analysis, there was no sister group of N. nuchalis ApoAI among the large clades of fish species. The transcriptional expression level of ApoAI was higher in N. nuchalis inhabiting the deep-bay area with a high salinity (over 31 psu) than in N. nuchalis inhabiting the top of the estuary with a low salinity (6~15 psu). In addition, the expression patterns of N. nuchalis ApoAI were positively correlated with environmental factors (transparency, pH, TC, and TIC) in the high salinity area. These results suggest that ApoAI gene expression can reflect habitat characteristics of N. nuchalis which traverse broad salinity ranges and is associated with functional roles of osmoregulation and lipid metabolism for fish growth and development

We believe that our findings would satisfy your journal requirements for the publication of this paper. Therefore, please be kind enough to evaluate our manuscript and inform us your comments. The English in the manuscript has been corrected by a professional science-editing service (Editage). In addition, our specific responses to the points raised by the reviewers are presented below.

Reviewer #2

Dear Authors,

Thanks for addressing most of my comments.

I found once again errors in some species name. Please be sure to write, also in the title the correct species name, according to zoological code; for example Nuchequula nuchalis (Temminck & Schlegel, 1845). From the second time you can use the shortened form: N.nuchalis. For example at lines 75-76 Engraulis encrasicolus is written wrongly.

Please check throughout the entire manuscript.

All the best regards

The Reviewer

  • In lines 3-4; we added zoological code as (Temminck & Schlegel,1845) in the title.
  • We revised some errors in line 16, lines 76-77, and line 283 as marked by yellow shades.

Thanks in advance.

Yours sincerely,

Ihn-Sil Kwak, Ph.D.

Dept. of Ocean Integrated Science,

Chonnam National University, Chonnam, 550-749, Korea.

E-mail address: iskwak@chonnam.ac.kr/inkwak@hotmail.com
